# Effects of Heat Treatment on Microstructure and Mechanical Properties of AlSi10Mg Fabricated by Selective Laser Melting Process

**Catherine Dolly Clement** [1,*] **, Julie Masson** [2] **and Abu Syed Kabir** [1]

1 Department of Mechanical and Aerospace Engineering, Carleton University, Ottawa, ON K1S 5B6, Canada; abusyedkabir@cunet.carleton.ca

2 Department of Aerospace Engineering, Institut Supérieur de l'Aéronautique et de l'Espace (ISAE-ENSMA), Poitou-Charentes, 31400 Toulouse, France; julie.masson@etu.isae-ensma.fr

* Correspondence: dollyclement@cmail.carleton.ca

**Abstract:** AlSi10Mg is the most widely additively manufactured and commercialized aluminum alloy and has been used in this study to analyze the effect of heat treatment on its microstructure and mechanical properties. Although research indicates AlSi10Mg parts produced by selective laser melting have characteristically very fine microstructures, there is a need for more intensive study to comprehend the effect of heat treatment on the mechanical properties of this alloy by analyzing its microstructure. In this study, AlSi10Mg specimens heat-treated at varying temperatures were analyzed by optical and electron microscopes. Micro-indentation hardness and tensile tests were performed to evaluate mechanical properties while considering the specimen build orientation. Observation shows that it is nearly impossible to completely dissolve the evolved second phase silicon-rich particles, which may have significant effects on the mechanical characteristics. Electron microscopy images show the evolution of iron-rich particles in the Al matrix, which may have a significant influence on the mechanical properties of the alloy.

**Keywords:** T6 heat treatment; 3D printing; additive manufacturing; selective laser melting; mechanical properties; tensile testing; hot cracking; microstructure; precipitation strengthening





## 1. Introduction

Selective laser melting (SLM) is one of the most popular additive manufacturing (AM) processes due to its ability to produce near-net-shaped metal parts with complex structures from a 3D model [1]. Although many different metallic systems have been printed successfully with SLM techniques, it is still quite a challenge to manufacture most of the aluminum alloys utilizing this technique, mostly because of the high reflectivity of the aluminum powder, the formation of oxide layers on top of the melt pools, and the wide solidification temperature range for the high strength Al-alloys [2]. High-strength aluminum alloys, (e.g., Al7075, Al2024) are extremely prone to hot cracking due to the presence of high alloying elements during solidification processes [3]. AlSi10Mg, on the other hand, shows excellent fusion weldability and castability and hence is the most additively manufactured aluminum alloy [2]. There have been several studies on the additively manufactured AlSi10Mg alloy focusing mostly on the microstructure [4,5], process parameters [6,7], mechanical properties [4,8], and influence of heat treatment [9].

Takata et al. [10] studied the changes in the microstructure of SLMed AlSi10Mg alloy after annealing at 300 °C for 2 h or solution treating at 530 °C for 6 h, reported fine Si-enriched precipitates within the columnar $\alpha$ and coarsening of Si particles which inhibits grain growth in the $\alpha$-Al matrix resulting in {001} texture. Li et al. [11] investigated the change in size and quantity of the eutectic Si particles upon heat treatment and found that when the solution was treated at 450 °C for 2 h, the average size of the Si particles

was less than 1 μm. However, with the increase in the solution treatment and artificial aging temperatures, the Si particles grew up to a size of 5 μm. This increase in the size of precipitates was accompanied by a decrease in their number due to particle coalescence and Ostwald's ripening phenomenon.

Research efforts on the optimization of AMed AlSi10Mg alloy include the microstructural development in the as-built state and after various heat treatments. The as-built AlSi10Mg alloy typically exhibits a so-called "fish-scale" pattern along the longitudinal direction and columnar pattern along with the melt pool formation in the transverse direction [9]. Increased strength values were observed in additively manufactured AlSi10Mg with a modified T5 treatment that exhibited a 64% increase in yield stress compared to as-built specimens [9] The conventional T6-like heat treatment is extensively used to increase the strength of cast Al-Si alloys with less attention provided to selectively laser melted aluminum alloys [4,12]. Aboulkhair et al. [4] observed an increase in hardness due to precipitation hardening after the aging of SLMed AlSi10Mg alloy compared to the solution heat treatment but also questioned whether $Mg_2Si$ is precipitated during heat treatment, which requires further research.

Alghamdi et al. [13] investigated the effects of the conventional heat treatment process and various cooling rates on the morphology and texture of Si precipitates and reported that the growth pattern of the eutectic-Si particle is greatly anisotropic. Wang et al. [12] performed a comprehensive study on the effects of the T6 heat treatment process on the densification, hardness, and tensile strength of AlSi10Mg stated that only limited resources are available in the study of heat treatment process specific to SLM-manufactured AlSi10Mg alloys and there is an increasing need to clarify the effects of T6 heat treatment process and its influence on mechanical properties in SLM-fabricated AlSi10Mg alloys.

Although the above-listed works on the effects of heat treatments provide us with extensive data, the need for more research is required to further tailor the heat treatment procedure to obtain the desired mechanical properties needed in operation [14] and not much focus is provided on the influence of thermal heat treatments on the microstructural behavior of additively manufactured AlSi10Mg [15]. Most authors have reported the temperatures for both solution and aging treatments ranging between 500–545 °C and 150–180 °C, respectively [11,15–17], and the peak hardening observed during the aging process is less frequently reported [18].

In the present work, we have adopted a T6 heat treatment procedure, which involved solution treating the specimens at 540 °C and artificially aging the samples at 140 °C and 160 °C. To the authors' knowledge, there have been no works reported so far in the literature at an aging temperature of 140 °C and very few works have been recorded for the solution treatment temperature of 540 °C. Additionally, this study also shows that the aging temperature can be tailored to as low as 140 °C to obtain maximum hardness, as most researchers used aging temperatures above 150 °C [11,17,19]. This case study was performed to provide a better emphasis on understanding the microstructure of heat-treated AlSi10Mg in comparison with as-built specimens and identify the peak hardening observed at the chosen optimized heat treatment conditions.

## 2. Materials and Methods

The AlSi10Mg powders with an average diameter between 21–27 μm were supplied by EOS. The detailed characterization of the powder was performed by Manfredi et al. [20]. AlSi10Mg dog bone specimens in Figure 1b were fabricated with dimensions as specified in Figure 1c, by selective laser melting process using an EOS M290 metal 3D printer. The specimens were built with build orientation indicated by the white arrow shown in Figure 1a. Using a low-density support structure, the build substrate was preheated to 200 °C.

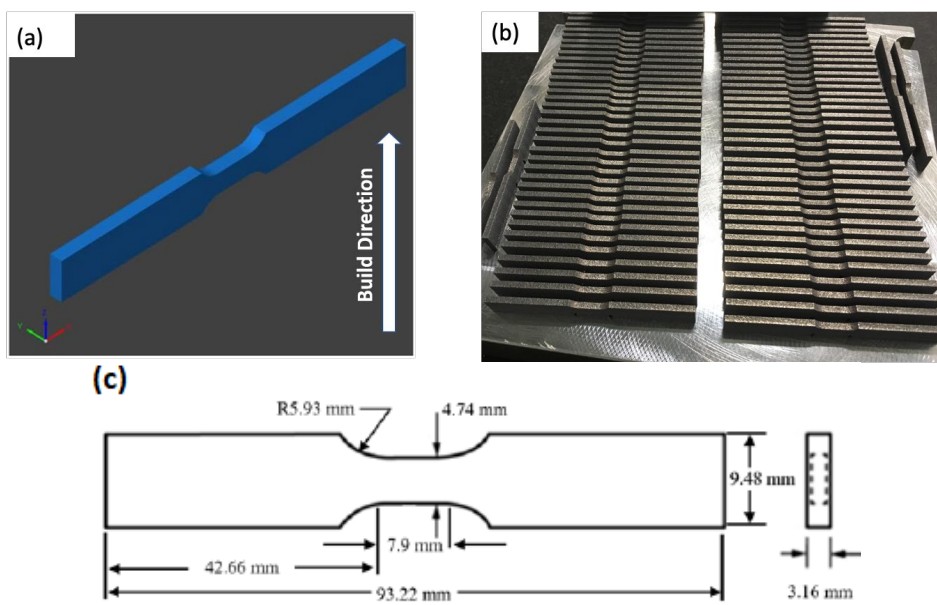

**Figure 1.** (**a**) Build orientation and, (**b**) as-built tensile coupons, (**c**) tensile specimen dimensions.

The chemical composition of the test specimens as provided by the supplier is given in Table 1. The optimized processing parameters utilized in fabricating 58 coupons employing the island scanning strategy are as follows: laser power 400 W, laser scan speed 7.0 m/s, and powder layer thickness 30 µm.

**Table 1.** Chemical composition of AlSi10Mg in wt% for this study.

| Al | Si | Fe | Cu | Mn | Mg | Ni | Zn | Pb | Sn | Ti |
|---|---|---|---|---|---|---|---|---|---|---|
| Bal. | 9.0–11.0 | <0.55 | <0.05 | <0.45 | 0.2–0.45 | <0.05 | <0.10 | <0.05 | <0.05 | <0.15 |

A conventional T6 post-build heat treatment was carried out for the coupons. Specimens were solution treated at 540 °C for 6 h followed by water quenching, this duration was chosen as most works were done at 1–2 h [11,14,16] and hardness has been found to increase with time due to solid solution strengthening that occurs when a supersaturated solid solution is developed during the solution treatment stage and preserved by rapid quenching. Further explanation on this is provided in the discussion section. The solution-treated samples were artificially aged at 140 °C and 160 °C for 1 to 16 h and water quenched to preserve the final microstructure.

The microstructures of the specimens were characterized by AmScope ME1200 series optical microscope and Tescan Vega-II XMU scanning electron microscope (SEM) equipped with an energy dispersive spectrometer (EDS). The longitudinal and transverse sections of the samples were mounted and metallographically prepared by grinding followed by fine polishing and etched with Keller's reagent.

The tensile strength of heat-treated AlSi10Mg specimens was characterized by an MTS universal testing machine at a constant strain rate of 2 mm/min. Vickers micro-indentation hardness was measured by Clemex microhardness tester at a load of 100 gm and a dwell time of 10 s.

## 3. Results and Discussion

Microstructures obtained by both optical and electron microscopes of the as-built AlSi10Mg are shown in Figure 2. The longitudinal section (Figure 2a,c) of the specimen shows typical fish scale morphology with a classic semi-circular profile with each layer made of melt pools defining the scan region, and it can also be seen that the overlapping of the melt pools leads to a partial re-melting of the material. This anisotropy is observed in the

shape of the melt pool, which is a result of high thermal gradients and a high cooling rate that contributes to the unexpected microstructure of additively manufactured aluminum alloys [21].

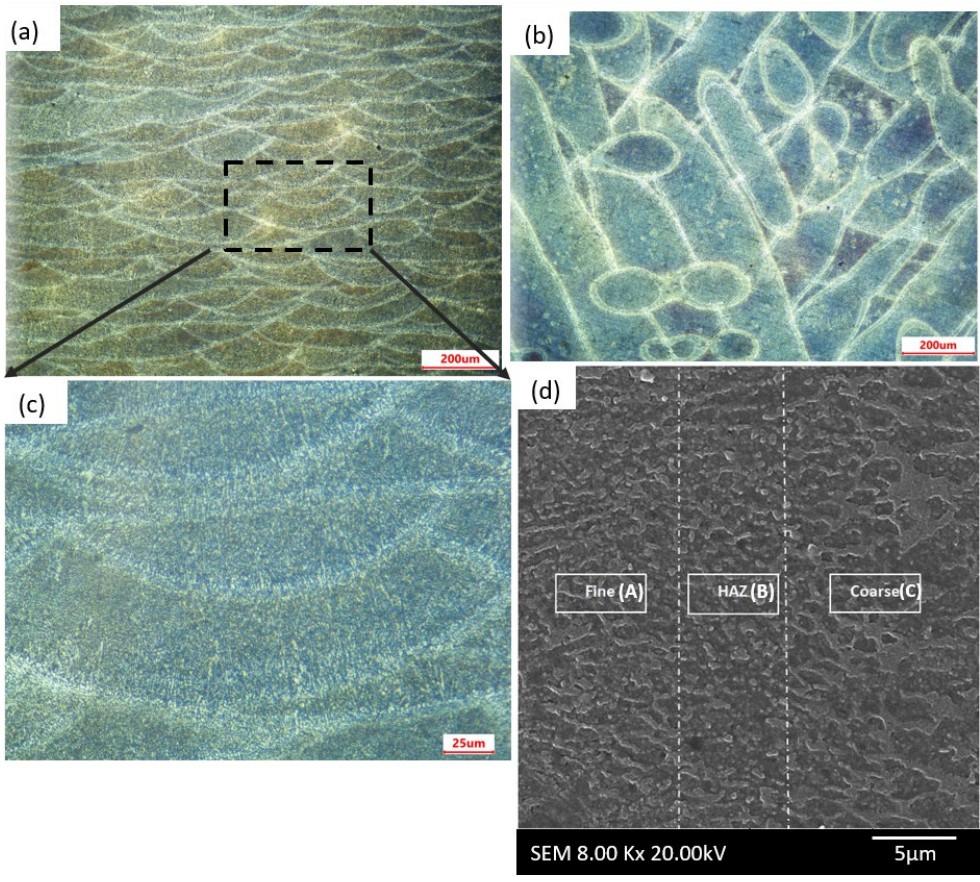

**Figure 2.** Optical micrographs of as-built AlSi10Mg alloy: (**a**) longitudinal section at low magnification; (**b**) transverse section; (**c**) longitudinal section at higher magnification; (**d**) SEM image of transverse section at higher magnification with heat-affected zones.

The transverse sectional view (Figure 2b) clearly shows the track segment morphology, from which the scan strategy can be ideally deduced to be made by a 67° rotation of the laser between each layer. The higher magnification SEM micrograph in (Figure 2d) reveals the microstructural evolution of the $\alpha$-Al matrix (dark phase) and the Si-rich continuous eutectic precipitates (bright phase). Three distinguishable zones are observed due to exposure to different temperature gradients across the melt track during the build process. The primary zone A is the interior region of the melt pool boundary, namely the melted zone, which is characterized by a fine microstructure. The secondary zone B represents the melt pool boundary with a relatively coarse microstructure and corresponds to the heat-affected zone. The tertiary zone C is characteristic of the re-melted layer enclosing a coarser microstructure. The formation of these distinct zones is attributed to differences in solidification rates in the molten pool area along the direction of the thermal gradient [22].

Microstructures of the solution-treated specimens at 540 °C for 6 h were observed under the optical microscope as provided in Figure 3, which shows a uniform microstructure completely devoid of the track segment morphology in both longitudinal and transverse sections. The uniform distribution of Si particles is clearly noticeable in the microstructure, mainly because of the spheroidizing of the eutectic phase during the heat treatment according to the Ostwald ripening mechanism [11]. The microstructure coarsened, causing Si to diffuse out to form second phase particles instead of clustering at grain boundaries of $\alpha$-Al. The highly saturated Al matrix causes excess Si to precipitate out, which is readily

influenced by the high temperature during the solution treatment. Upon solution treatment, the continuous network of eutectic Si walls is broken down, ultimately causing residual Si atoms dissolved in the Al matrix to be precipitated out after quenching [13].

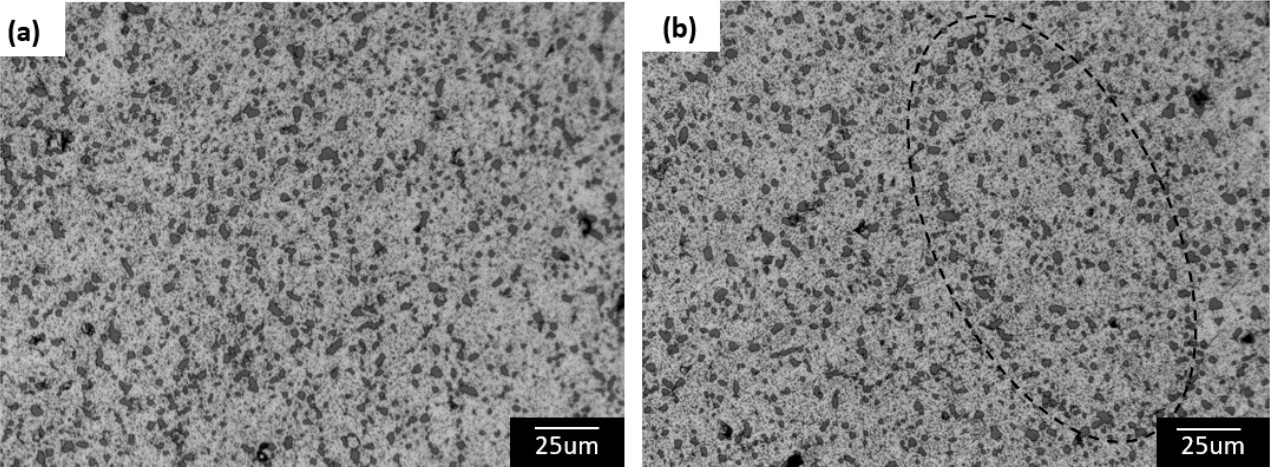

**Figure 3.** Optical micrographs of specimen ST for 6 h; (**a**) longitudinal section; (**b**) transverse section.

The dotted circle in Figure 3b is indicative of Si particles precipitating along the melt pool boundaries. This is coherent because the regions lie at the overlap of two hatches that are remelted regularly during the processing. It also shows that even though the growth of the second phase particle is preferentially along the melt pool boundaries, they are also uniformly distributed in the Al matrix.

To investigate the distribution of Al and other alloying elements, EDS area mapping was performed on the solution heat-treated specimens. The BSE image and its corresponding elemental maps of Fe, Si, and Mg are shown in Figure 4. It is inferred from the elemental maps that Si is locally distributed, and Mg is uniformly distributed in the matrix. The elemental analysis also identified the slender needle-shaped brighter precipitates with a high concentration of Fe (marked by the arrowhead) as shown in the results in Figure 4a. Spot analysis on these brighter precipitates as shown in Figure 4e also indicates the presence of Fe in them. Intermetallic particles form due to the presence of Fe in the alloy's chemical composition as reported in Table 1. Heat treatment induces diffusion and segregation of Fe and Si atoms leading to the formation of Fe-rich precipitates and random distribution of Si particles in the aluminum matrix [23].

The Vickers micro-indentation hardness curves obtained after artificial aging at 140 °C and 160 °C for 1 to 16 h are shown in Figure 5. Hardness drops about 25% after solution treatment from the as-built condition due to the significant softening of material caused by microstructure coarsening. Hardness increases with aging time and the peak hardness were recorded at 8 h for the 160 °C and 12 h for 140 °C. The increase in hardness with time could be attributed to both solid solution and precipitation strengthening. Solution treatment at 540 °C results in a supersaturated solid solution existing at room temperature followed by aging at 140 °C and 160 °C contributes to accelerated precipitation of Si atoms into the Al matrix. The significant difference in size between the Al atoms and the Si solutes produces a greater disruption of the initial crystal structure, making dislocation slip more difficult and thereby increasing the strengthening effect. Although upon heat treatment, the microstructure is no longer ultrafine-grained, which reduces the contribution of grain boundaries to the strengthening effect. With increasing Si precipitating into the Al matrix, the dislocation motion is obstructed by Si dispersoids [14].

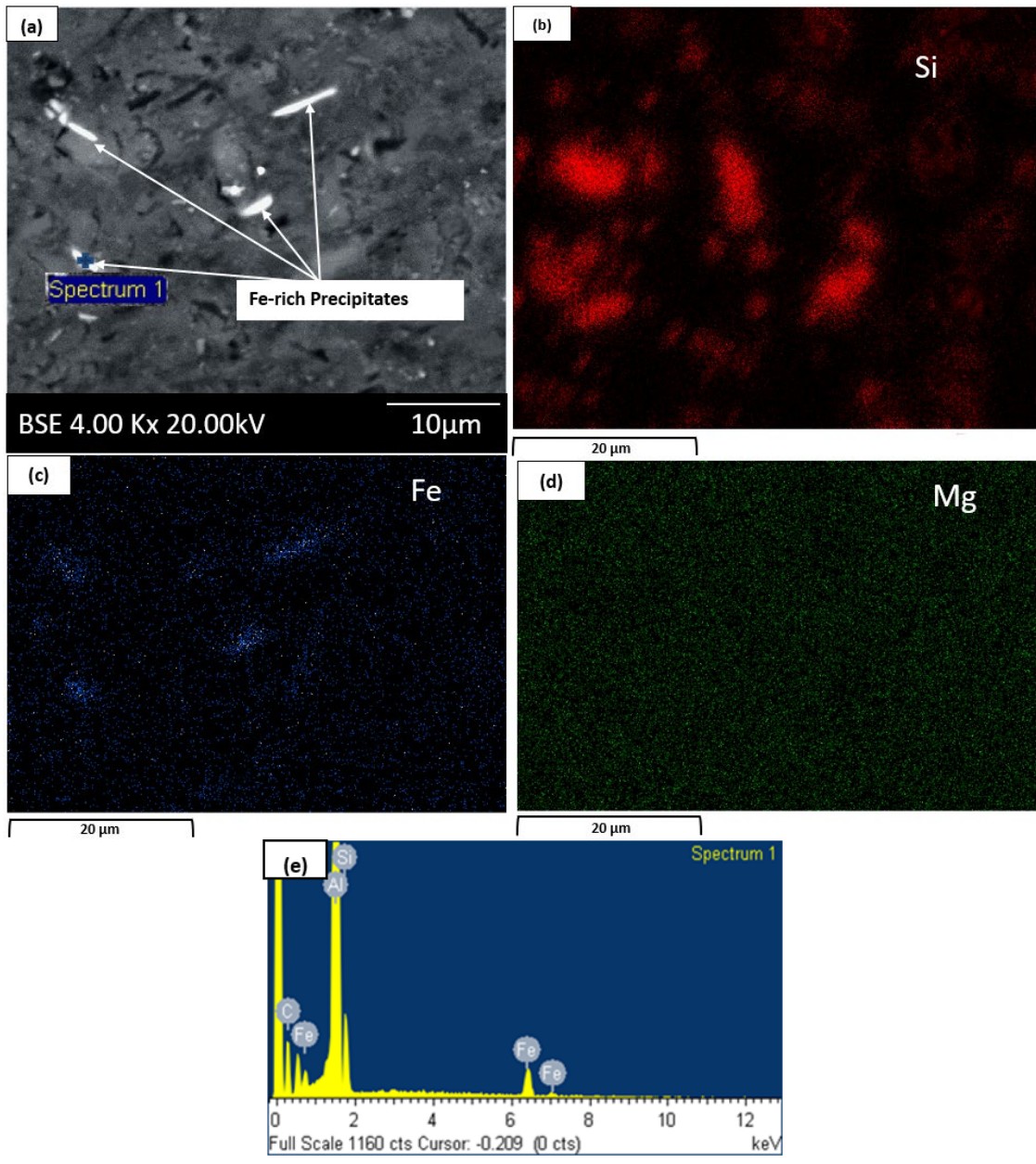

**Figure 4.** EDS analysis of specimens ST at 540 °C for 6 h. (**a**) SEM image of Al-Si-Fe precipitate and corresponding EDS maps of (**b**) Si; (**c**) Fe; (**d**) Mg (**e**) spot analysis spectrum.

It can be inferred from the hardness curve that the maximum hardness is obtained at 12 h and 8 h for 140 °C and 160 °C, respectively. However, it is efficient to perform heat treatment at 160 °C as it will promote the achievement of desirable hardness results at a shorter duration compared to performing heat treatment at 140 °C. Hardness goes down after these optimized hours due to over-aging as the size of the precipitates grows bigger and the dislocations–precipitation interaction changes. When the precipitate size was smaller, the dislocations were blocked by the Orowan looping mechanism, whereas, during over-aging, dislocations were able to bypass the larger precipitates.

There are no obvious differences observed between the microstructure of the SHT specimens and the AA specimens from the optical micrographs in Figure 6, although it is important to note that there is still the existence of Si particles in the Al matrix, and it is quite intuitive that there are slight variations in the density and size of the Si particles. Most Si particles were observed to be irregular in shape and there were no apparent differences

in shape and density of Si particles after SHT. However, the Si particles in aged specimens appear nearly spherical as seen in the electron micrographs in Figure 7b,c. Eutectic Silicon spheroidizes at elevated temperatures due to the increase in size and surface perturbations occurring at the interface, resulting in the formation of near-spherical Si particles. The average volume percent of Si precipitates and the size of the particles are calculated to be 22% and 1.32 μm, respectively, using ImageJ software. The average volume percent of Fe-rich precipitates in SHT and aged specimens was calculated to be 5.9% and 6.12%, respectively, which may contribute to the enhanced mechanical properties of the alloy.

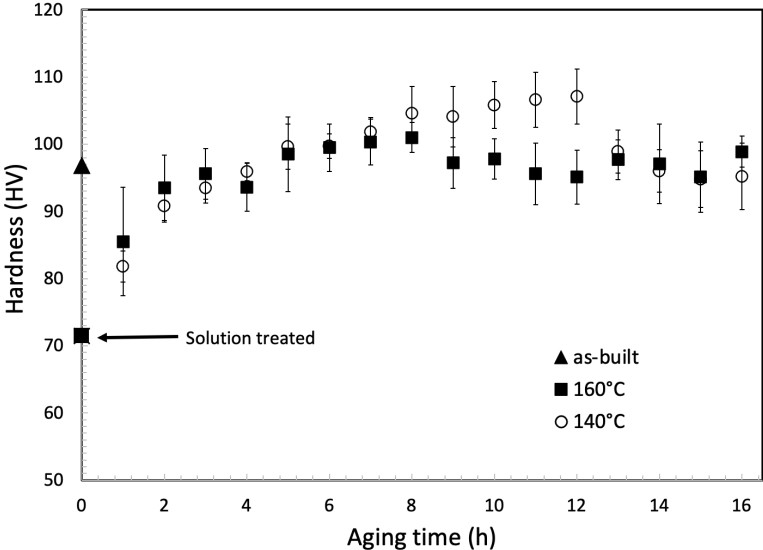

**Figure 5.** Hardness testing curves obtained for specimens for as-built and heat-treated specimens.

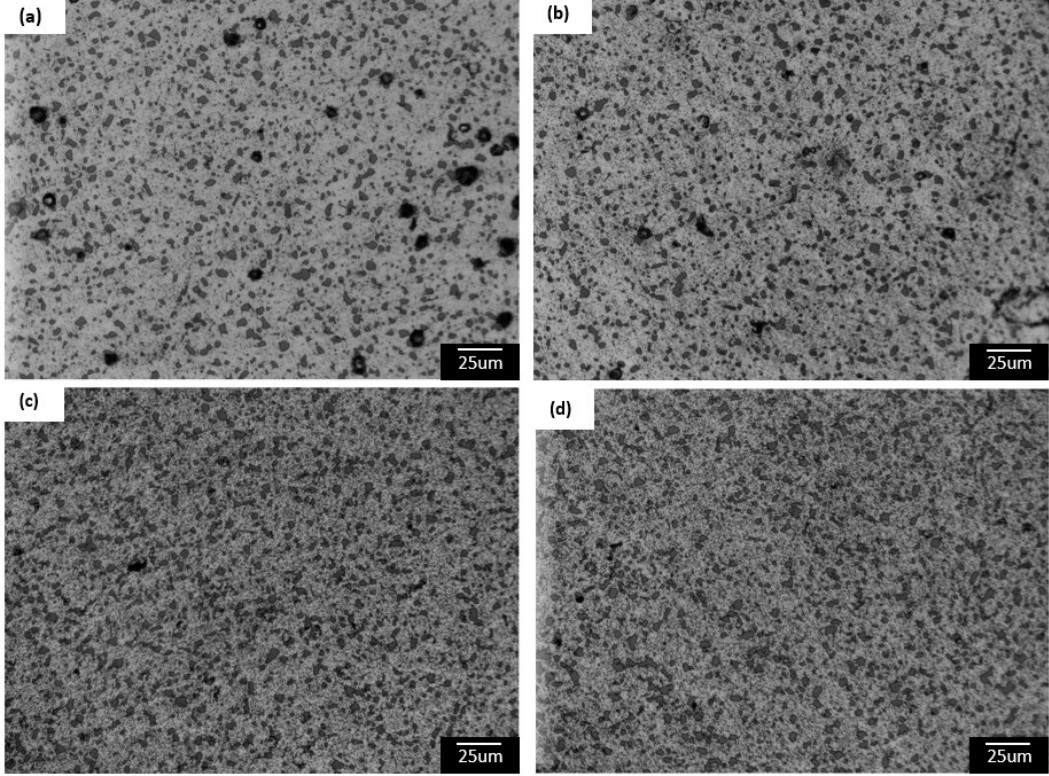

**Figure 6.** Optical micrographs of specimens artificially aged at (**a**) 140 °C and 11 h (**b**) 140 °C and 12 h (**c**) 160 °C and 7 h (**d**) 160 °C and 8 h.

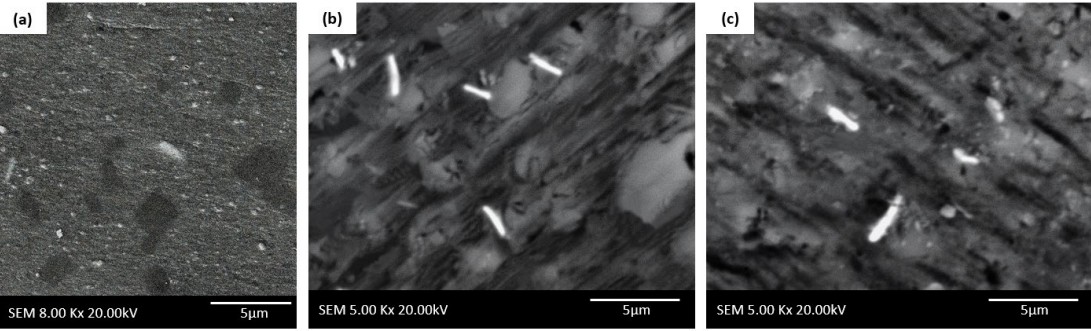

**Figure 7.** Electron micrographs of specimens ST at (**a**) 540 °C for 6 h and AA at (**b**) 140 °C for 12 h (**c**) 160 °C for 8 h.

Room temperature tensile testing was performed on the as-built and heat-treated specimens and the corresponding engineering stress–strain curves and tensile properties are provided in Figure 8 and Table 2, respectively. The mechanical properties of AMed AlSi10Mg are often compared to the as-cast A360 alloy due to the similar composition and they both constitute an Al-Si system with microstructures containing α-Al grains in a eutectic Al-Si matrix [24]. In this study, the tensile strength of SLMed AlSi10Mg exhibits a 21.3% increase in yield strength and a 54% increase in elongation at break with a slight decrease in UTS compared to the conventional as-cast A360 tensile strength of 317 MPa [25]. Tensile strength after solution treatment was significantly reduced but the ductility was improved at the same time. Strength decreases upon heat treatment as the Si atoms trapped in the Al matrix are rapidly precipitated out onto the existing eutectic network Si, thus increasing the size of Si precipitates and promoting dislocation bypass in contrast to grain boundary strengthening in as-built specimens. However, the improvement of ductility can be explained by the dependency of tensile ductility on grain orientation in the heat-treated samples [26]. This is also in agreement with studies by Han and Jiao [22] that showed a substantial increase in elongation to fracture from 6 to 22% due to grain growth associated with Si precipitation, and the thermal residual stress release that occurred during heat treatment. Surprisingly, the yield strength of the AA (160 °C) samples shows an 11.4% increase compared to that of AA (140 °C). This contradicts the hardness data that showed a slight increase at the peak for AA (140 °C) samples. A similar trend was observed in additively manufactured AlSi10Mg by Girelli et al. [27] where lower hardness values were recorded at higher aging temperatures and were correlated to the presence of porosity defects. It was stated that aging at higher temperatures allows the diffusion of gas atoms inducing non-uniform distribution of porosities, which results in such anomalies.

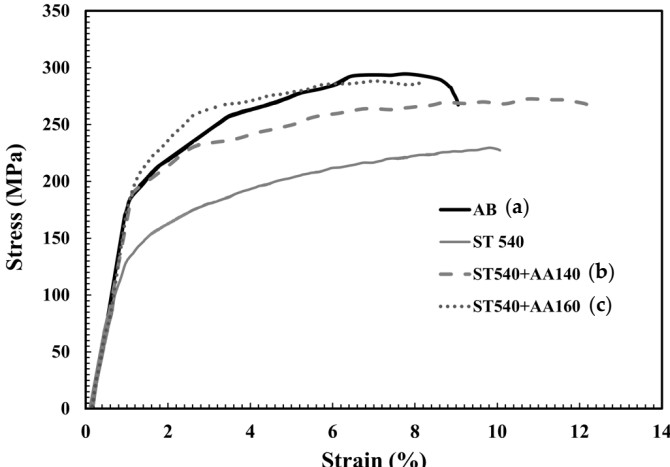

**Figure 8.** Stress–strain curves, YS, UTS and Elongation at break for (**a**) as-built (**b**) ST + AA at 140 °C for 12 h (**c**) ST + AA at 160 °C for 8 h.

**Table 2.** Tensile Properties of AlSi10Mg Specimens.

| | YS (MPa) (0.2% Offset) | UTS (MPa) | Elongation at Break |
|---|---|---|---|
| **As-built** | 200 | 293 | 7.6% |
| **ST 540 °C** | 130 | 217 | 8.8% |
| **ST 540 °C + AA 140 °C (6 h) (12 h)** | 195 | 269 | 10.8% |
| **ST 540 °C + AA 160 °C (6 h) (8 h)** | 220 | 280 | 6.4% |

Fracture surfaces of the as-built and heat-treated specimens at different magnifications are provided in Figure 9. The as-built samples exhibit quite a different morphology compared to the heat-treated specimens, with a smooth dimple surface and "river like" pattern around the ridges, which is a characteristic of short and discontinuous track segments, and the existence of partially melted powder particles results in large cracks contributing to brittle fracture and low ductility (Figure 9a). A previous study [21] showed that eutectic Si precipitated in as-built specimens act as crack initiation sites resulting in crack propagation that is detrimental to ductility in contrast to Si precipitated in heat-treated specimens that attain uniform distribution over the Al matrix.

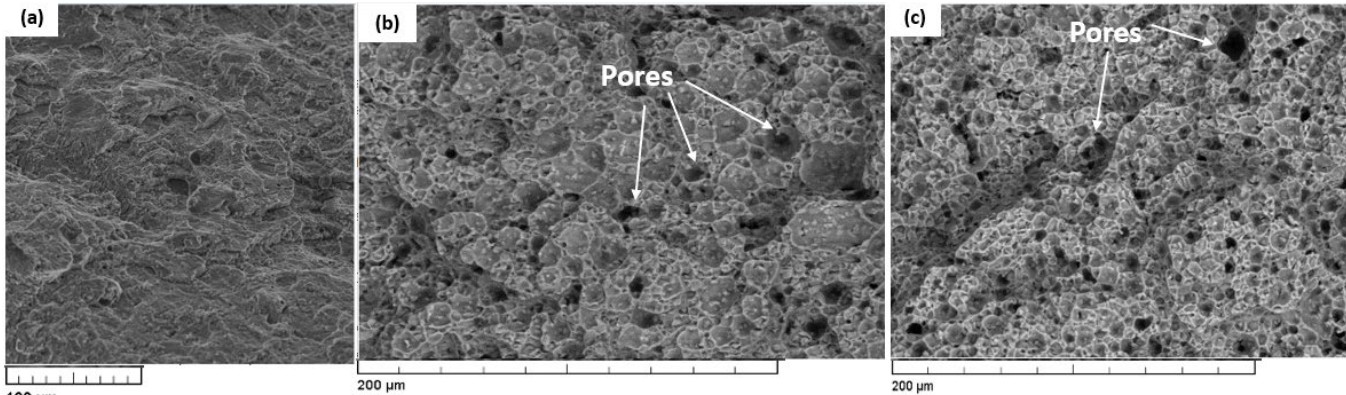

**Figure 9.** SEM images of fracture morphology of specimens (**a**) as-built (**b**) ST + AA at 140 °C and 12 h (**c**) ST + AA at 160 °C and 8 h.

Both artificially aged samples showed similar fracture surfaces with dimple structures, indicating the presence of micro-voids to be the initiator of ductile failure as shown in Figure 9b,c. However, the significant differences in ductility values could be explained due to the distribution and quantities of pores, which appear much higher in the AA 160 °C samples thereby impacting the ductility of the specimen [19]. Segregation of Si particles into randomly distributed spheroids where voids can nucleate promoting void coalescence and thereby leading to crack propagation resulted in specimen failure under tensile loading in heat-treated specimens [4].

## 4. Conclusions

In this research, we concentrated on the effects of T6-like heat treatment on the microstructural behavior and mechanical properties of additively manufactured AlSi10Mg alloy. Some of the important observations made are as follows.

- The microstructure of as-built AlSi10Mg shows a distinct morphology between different sections with a characteristic fish scale pattern for longitudinal sections and a clustered columnar pattern for the transverse sections.
- The inhomogeneity of the melt pool results in a range of varying mechanical properties with a need to homogenize the microstructure to obtain better control over the mechanical properties.

- Though the heat-treated specimens exhibit a homogenized microstructure, it is impossible to completely eliminate all the precipitates. Artificially aged specimens provided maximum hardness at 12 h for 140 °C and 8 h for 160 °C. This represents a 10% increase in hardness values obtained for heat-treated samples compared to as-built specimens due to precipitation hardening. Despite a little increase in the volume percent of Si-rich precipitates and precipitation coarsening, the increase in hardness value may be associated with the increasing Fe-rich precipitates.
- Tensile strength and elongation at the break of the as-built SLM'ed AlSi10Mg are comparatively higher than that of conventional cast A360 aluminum alloy, although the ultimate tensile strength is slightly lower. This is due to the lower concentration of Mg to form the $Mg_2Si$ complex in AlSi10Mg, which is a major contributor to precipitation hardening for A360. The imposition of heat treatment at chosen optimal conditions providing peak hardening induces an increase in yield strength and elongation at break and a decrease in UTS.
- Although higher hardness was obtained after artificial aging at 140 °C compared to 160 °C, tensile strength was found to be higher at 160 °C. Tensile elongation, on the other hand, was significantly higher for the 140 °C specimen. This anomaly was mainly due to the large amount of micro-porosity found in the fracture surface at 160 °C.

This work provides an overall assessment of microstructural characterization and mechanical properties of AlSi10Mg produced by selective laser melting process and identified the impact of T6-like conventional heat treatment. The results obtained could be used as a manufacturability reference to produce SLM'ed AlSi10Mg at optimized parameters, and to obtain better control of the microstructural and mechanical properties.

**Author Contributions:** C.D.C.: conceptualization, methodology, writing—original draft preparation, and visualization; J.M.: writing—review and editing; A.S.K.: conceptualization, writing—review and editing, supervision and resources. All authors have read and agreed to the published version of the manuscript.

**Funding:** This research received no external funding.

**Institutional Review Board Statement:** Not Applicable.

**Informed Consent Statement:** Not Applicable.

**Data Availability Statement:** The data presented in this study are available on request from the corresponding author.

**Acknowledgments:** Authors wish to acknowledge the Natural Sciences and Engineering Research Council (NSERC) Canada for the financial support and Peter Walker from Carleton University for support in metallography and SEM analysis.

**Conflicts of Interest:** The authors declare no conflict of interest.

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
