# Peer review of "Effects of Heat Treatment on Microstructure and Mechanical Properties of AlSi10Mg Fabricated by Selective Laser Melting Process"

_jmmp, doi:10.3390/jmmp6030052_

Round 1

Reviewer 1 Report

Article of title: "Effects of Heat Treatment on Microstructure & Mechanical Properties of AlSi10Mg Fabricated by Selective Laser Melting Process" by Authors: Catherine Dolly Clement, Julie Masson, Abu Syed Kabir, is written correctly.

The impact of heat treatment on the mechanical properties of this alloy by analyzing its microstructure  is presented in the manuscript.

The article AlSi10Mg specimens heat treated at varying temperatures were analyzed by optical and electron microscopes was presented.

The article is well written, however, it must be corrected before publication.

          In Table 2. "Tensile Properties of AlSi10Mg Specimens" does not meet the editorial requirements, units must be only on one line. The graphical form does not look good, and the units are not entered in the values.

          On Figure 9. "SEM images of fracture morphology of specimens (a) as-built (b) ST+AA at 140°C& 12h (b) 270 ST+AA at 160°C & 8h", there are signs "pores" on the basis of which this was confirmed, they may also be "hills".

           In chapter "4. Conclusions" I propose to clearly list the most important statements and conclusions in sub-items or from dashes.

Author Response

1. In Table 2. "Tensile Properties of AlSi10Mg Specimens" does not meet the editorial requirements, units must be only on one line. The graphical form does not look good, and the units are not entered in the values.

A : Per the reviewer’s suggestions we have removed the units from within the table and is now included only in the column Title. The graphs have been plotted as per the journal standards and the units for the values in the graph are included in the axis title.

2. On Figure 9. "SEM images of fracture morphology of specimens (a) as-built (b) ST+AA at 140°C& 12h (b) 270 ST+AA at 160°C & 8h", there are signs "pores" on the basis of which this was confirmed, they may also be "hills".

A : In SEM, the depth of focus is high enough to differentiate between a pore and a hill. The authors are sure about the fact that the arrow indicated entities shown in the image are pores. Also, in comparison with the SEM image of as-built specimen the darker areas identified as pores in our work has considerably increased in the SHT and AA specimens which is directly induced by the diffusion of gas atoms at higher temperatures which would unlikely be the case with hills. Additionally, the references (15&23) used in discussion identify similar entities in their results as pores.

3. In chapter "4. Conclusions" I propose to clearly list the most important statements and conclusions in sub-items or from dashes.

A : Conclusion has now been re-written in bullet form.

Reviewer 2 Report

This paper focused on the Effects of Heat Treatment on Microstructure & Mechanical Properties of AlSi10Mg Fabricated by Selective Laser Melting Process. The article is well organized. Moreover, good results are discussed about the effect of heat treatment on this alloy's microstructure, mechanical properties, and precipitation behavior. It can be published in this journal after some changes have been made.

  1. The literature review should be recast to highlight the information gap correctly.
  2. The novelty of the present study should be clearly mentioned in the introduction.
  3. The relationship between microstructure and mechanical properties should be discussed in the manuscript.
  4. Why highly saturated Al matrix causes excess Si to precipitate out, which is readily influenced by the high temperature during solution.
  5. The author should explain the differences between the SHT specimens and the AA specimens regarding microstructure and mechanical properties. Additionally, the author should explain the differences between morphology, size, type, and density of Si particles in both conditions.
  6. The author should explain the main reason for higher hardness and tensile strength after aging at 140°C and 160°C, respectively, and correlate it with microstructure.
  7. The author should need to rewrite the conclusions part more clearly.
  8. In some of the figures, texts figures are not readable. Improve figure quality.

Author Response

1.The literature review should be recast to highlight the information gap correctly.

A:  The following paragraph has been added to the literature review:

Alghamdi et al.[10] investigated the effects of conventional heat treatment process and various cooling rates on the morphology and texture of Si precipitates and reported that the growth pattern of eutectic-Si particle is greatly anisotropic. Wang et al.[9] performed a comprehensive study on the effects of T6 heat treatment process on the densification, hardness and tensile strength of AlSi10Mg and stated that only limited resources are available on the study of heat treatment process specific to SLM-manufactured AlSi10Mg alloys and there is an increasing need to clarify the effects of T6 heat treatment process and its influence on mechanical properties in SLM-fabricated AlSi10Mg alloys.

2. The novelty of the present study should be clearly mentioned in the introduction.

A : The work is novel to the best of our knowledge in such a way that there has been no works reported so far in the literatures at an aging temperature of 140°C and very few works have been recorded for the solution treatment temperature of 540°C. Additionally this study also shows that the aging temperature can be tailored to as low as 140°C to obtain maximum hardness as most researchers used aging temperatures above 150°C, which is mentioned in paragraph 6 in introduction.

3. The relationship between microstructure and mechanical properties should be discussed in the manuscript.

A:  Pages 7 through 12 in the results and discussion section provide detailed reports on the relationship between microstructure and its influence on hardness and tensile properties. This is one of major parts in the results and discussion part.

4. Why highly saturated Al matrix causes excess Si to precipitate out, which is readily influenced by the high temperature during solution 

A:  Added “Upon solution treatment, the continuous network of eutectic Si is broken down causing residual Si atoms in the Al matrix to be precipitated out”

5. The author should explain the differences between the SHT specimens and the AA specimens regarding microstructure and mechanical properties. Additionally, the author should explain the differences between morphology, size, type, and density of Si particles in both conditions.

A:  Paragraph 8 in the results & discussion section on page 9 talks about the observation in differences in microstructure of SHT and AA specimens. Additionally, distribution of Si and Fe precipitates were also quantified in the same paragraph.

6.The author should explain the main reason for higher hardness and tensile strength after aging at 140°C and 160°C, respectively, and correlate it with microstructure

A: These are explained in detail in the results & discussion section on page 7 & 10. Here are some of the details re-quoted from those pages:

Artificially aged specimens provided maximum hardness at 12h for 140°C and at 8h for160°C. This represents a 10% increase in hardness values obtained for heat treated samples compared to as-built specimens due to precipitation hardening. The significant differences in ductility values could be explained due to the distribution and quantities of pores which appear much higher in the AA 160°C samples thereby impacting the ductility of the specimen.

7. The author should need to rewrite the conclusions part more clearly.

A:  Conclusion has been broken down into bullet points to better aid in clarity of information presented.

8. In some of the figures, texts figures are not readable. Improve figure quality

A:  Fig 2, Fig 3, Fig 6, Fig 7 have been formatted to improve visual quality and texts within have been replaced with manually created data.

Reviewer 3 Report

This article is devoted to the study of objects synthesized from AlSi10Mg by selective laser melting. The novelty of the study lies in the use of new modes of heat treatment that have not been previously described in the literature. In general, the manuscript leaves a positive impression and contains all the necessary stages of research work. However, there are a few comments that need to be taken into account before publication.
1. A large number of articles on the AlSi10Mg alloy have been published to date. At the same time, the list of used sources contains only 23 references. I suggest that the authors cover the previous studies in more detail.
2. Parameters of selective laser melting. How were these parameters selected? If based on the analysis of the literature, then appropriate references should be provided.
3. Was the platform heating used in the synthesis of samples?
4. Table 2. Include the level of properties from previously published studies in the table. You can also add the properties of aluminum-based composite materials. This way the article will be more interesting and informative for readers. For example, there are several sources that can be used:
DOI: 10.1016/j.jmst.2018.09.004
DOI: 10.3390/ma11030392
DOI: 10.1016/j.msea.2018.08.074
DOI: 10.3390/ma14102648
DOI: 10.1016/j.msea.2015.12.101

Author Response

1. A large number of articles on the AlSi10Mg alloy have been published to date. At the same time, the list of used sources contains only 23 references. I suggest that the authors cover the previous studies in more detail.

A:  Added references 5(DOI: 10.1016/j.jmst.2018.09.004), 7 (DOI: 10.1016/j.msea.2015.12.101), 8 (DOI: 10.1016/j.msea.2018.08.074), 13 (DOI : 10.1016/j.msea.2020.139296)  to the literature review.

2. Parameters of selective laser melting. How were these parameters selected? If based on the analysis of the literature, then appropriate references should be provided.

A:  Samples were fabricated with SLM parameters as suggested by the manufacturer (Burloak inc.) for the nature of our research. However, since the study does not focus on the effects of printing parameters on the microstructure and mechanical characteristics, we have not included detailed references in the literature review. The parameters were chosen by the experts at Burloak with only one objective, to produce sound quality tensile coupons.

3. Was the platform heating used in the synthesis of samples?

A: The build substrate was pre-heated to 200°C as mentioned in materials and methods section of the manuscript.

4. Table 2. Include the level of properties from previously published studies in the table. You can also add the properties of aluminum-based composite materials. This way the article will be more interesting and informative for readers. 

A: Ref  22, 24, 25, 26, 27 used in the discussion section on page 10 in the tensile characteristics section provide comparison of the tensile behaviour of AlSi10Mg with cast counterparts and similar additively manufactured aluminum alloys with varying heat treatment parameters. We have decided not to include aluminum-based composites to the discussion as they are not a suitable comparison to additively manufactured Al alloys due to vast differences in composition and microstructural behaviour.

Round 2

Reviewer 2 Report

The revised manuscript title “Effects of Heat Treatment on Microstructure & Mechanical Properties of AlSi10Mg Fabricated by Selective Laser Melting Process” should be accepted for publication in the present form.  

Author Response

Response : There are no revisions requested here and the present version of the paper is acceptable as noted by the reviewer.